# Structural elucidation of the haptoglobin–hemoglobin clearance mechanism by macrophage scavenger receptor CD163

Ching-Shin Huang[1]*, Hui Wang[2], Joshua B. R. White[3], Oksana Degtjarik[3], Cindy Huynh[2], Kristoffer Brannstrom[2], Mark T. Horn[2], Stephen P. Muench[4], William S. Somers[1], Javier Chaparro-Riggers[2], Laura Lin[1], Lidia Mosyak[1]*

**1** BioMedicine Design, Pfizer, Inc., Cambridge, Massachusetts, United States of America, **2** BioMedicine Design, Pfizer, Inc., La Jolla, California, United States of America, **3** School of Molecular and Cellular Biology, Faculty of Biological Sciences & Astbury Centre for Structural Molecular Biology, University of Leeds, Leeds, United Kingdom, **4** School of Biomedical Sciences, Faculty of Biological Sciences & Astbury Centre for Structural Molecular Biology, University of Leeds, Leeds, United Kingdom

* Ching-Shin.Huang@pfizer.com (C-SH); Lidia.Moysak@pfizer.com (LM)

## Abstract

Intravascular hemolysis releases hemoglobin into the bloodstream, which can damage vascular and renal tissues due to its oxidative nature. Circulating haptoglobin acts as a primary defense by binding to free hemoglobin, forming a haptoglobin–hemoglobin (HpHb) complex that is then recognized and cleared by the CD163 scavenger receptor on macrophages. While the function and structure of HpHb complex are mostly well-defined, the molecular mechanism underlying its interaction with CD163 remains unclear. Here we report the cryo-electron microscopy structures of human CD163 in its unliganded state and in its complex with HpHb. These structures reveal that CD163 functions as a trimer, forming a composite binding site at its center for one protomer of the dimeric HpHb, resulting in a 3:1 binding stoichiometry. In the unliganded state, CD163 can also form a trimer, but in an autoinhibitory configuration that occludes the ligand binding site. Widespread electrostatic interactions mediated by calcium ions are pivotal in both pre-ligand and ligand-bound receptor assemblies. This calcium-dependent mechanism enables CD163/HpHb complexes to assemble and, once internalized, disassemble into individual components upon reaching the endosome, where low calcium and lower pH conditions prevail. Collectively, this study elucidates the molecular mechanism by which CD163-mediated endocytosis efficiently clears different isoforms of HpHb.

## Introduction

When hemoglobin (Hb) is released during intravascular hemolysis, it poses a risk because the heme component can participate in chemical reactions that produce free radicals. To counteract elevated levels of free Hb in the plasma, the human body

**Data availability statement:** The cryo-EM density maps were deposited with the Electron Microscopy Database (EMDB) under accession codes EMD-49208, EMD-49209, EMD-49210, EMD-49211, EMD-49212, EMD-49213, EMD-49214, EMD-49215, EMD-49216, EMD-49217, EMD-49218, EMD-49219, EMD-49220 and EMD-49221. The structure coordinates were deposited with the Protein Data Bank (PDB) under accession code 9NB5, 9NB6, and 9NB8.

**Funding:** Funding for this research was provided by Pfizer, Inc., Cambridge, MA, USA. The authors are current or former employees of Pfizer, Inc., and were responsible for the study design, data collection and analysis, decision to publish, and preparation of the manuscript.

**Competing interests:** I have read the journal's policy and the authors of this manuscript have the following competing interests: C.-S.H., H.W., C.H., K.B., M.T.H, W.S.S., J.C.-R., L.L. and L.M. were full-time employees and shareholders of Pfizer at the time the research presented here was conducted.

**Abbreviations:** CCP, complement control protein; cryo-EM, cryo-electron microscopy; ECD, extracellular domain; Hb, hemoglobin; Hp, haptoglobin; HpHb, haptoglobin–hemoglobin complex; SEC-MALS, size exclusion chromatography coupled with multiple-angle light scattering; SP, serine protease; SPR, surface plasmon resonance; SRCR, scavenger receptor cysteine rich.

utilizes haptoglobin (Hp), a hemoglobin scavenger that circulates the bloodstream to find and tightly bind free Hb [1]. This binding confines the free Hb within the vascular system, preventing kidney damage and vascular injury. Furthermore, the complexation of Hp and Hb facilitates its recognition by CD163 receptors on macrophages, enabling degradation of the HpHb complex through endocytosis and subsequent heme metabolism within the cells [1,2].

Hp, serving as a first responder to detoxify free Hb in plasma, has been studied extensively [3,4]. In humans, there are two genetic variants of Hp: Hp1 and Hp2. The Hp1 variant consists of a complement control protein (CCP) domain followed by a serine protease (SP) domain. In contrast, the Hp2 variant has an additional CCP domain at the N-terminus due to gene duplication events [5]. The two variants lead to three Hp phenotypes—Hp (1–1), Hp (2–1), and Hp (2–2)—each arising from the CCP domain swapping that is stabilized by disulfide bond linkages [6,7]. Consequently, Hp (1–1) forms a simple homodimer, whereas Hp (2–1) and Hp (2–2) appear as a range of Hp multimers [2,6]. The structure of the dimeric Hp (1–1) complexed with Hb offered valuable insights into the initial step of Hb clearance [7], demonstrating that the SP domain of Hp effectively interacts with Hb preventing its radical propagation.

CD163 belongs to the scavenger receptor cysteine rich (SRCR) family and is a type I membrane protein with a short cytoplasmic tail, a single transmembrane helix, and a large ectodomain consisting of nine class B SRCR domains [8,9]. Three splice variants with different cytoplasmic tail lengths have been reported, with the short-tailed isoform having dominant expression [8]. The cytoplasmic tail, which includes a YREM motif recognized by the adaptor protein complexes for endocytosis [10–12], also has phosphorylation sites for casein kinase II and protein kinase C [13], although CD163 phosphorylation is reportedly not involved in Hb endocytosis [12]. On a structural level, the signature SRCR domain is a conserved fold of about 100 amino acids containing acidic residue clusters for metal ion binding and ligand recognition [14,15].

Early studies showed that the interaction between HpHb and CD163 is $Ca^{+2}$-dependent and pH-sensitive [2,16]. It was later revealed that the acidic clusters in the SRCR domain 2 and 3 of CD163 and two basic Arg/Lys residues in the loop of the Hp SP domain are essential for the interaction and a working model of $Ca^{+2}$-dependent electrostatic pairing for the receptor/ligand interaction was proposed [17–19]. A separate study demonstrated that Hb can interact with CD163 and can be internalized to endosomes for degradation in the absence of Hp [20]. Additionally, small-angle X-ray scattering measurements using truncated CD163 (SRCR domains 1–5) and dimeric Hp(1–1)Hb complexes suggested that Hp(1–1)Hb binds to two chains of CD163 [7]. Together, these reports indicated that CD163 interacts with HpHb through multiple binding sites. However, how these multiple sites function together in the context of CD163/HpHb assembly remained unknown due to the lack of structural information of an intact complex.

In this study, we utilized cryo-electron microscopy (cryo-EM) single particle reconstruction to determine the structures of CD163 in its unliganded state and in its complex with ligand Hp(1–1)Hb. The resulting two structures, with overall resolutions of 3.0 and 3.3 Å respectively, revealed that (1) CD163 functions as a trimer, enclosing

just one Hp(1–1)Hb protomer at its center, with a 3:1 receptor-ligand stoichiometry, (2) the unliganded CD163 can also form a trimer but displays an autoinhibitory structure, and (3) $Ca^{+2}$-coordinated electrostatic interactions create extensive contacts between HpHb and CD163 and among the three subunits of CD163, thereby nucleating and stabilizing this large assembly. Collectively, these findings provide an important structural framework for understanding the clearance mechanism of HpHb by CD163.

## Results

### CD163 self-assembles through $Ca^{+2}$-dependent electrostatic interactions

In our attempt to reconstruct the CD163-Hp(1–1)Hb complex, we produced the complete CD163 extracellular domain (ECD), encompassing the SRCR domains D1–D9, and first analyzed its structure in the unliganded state through cryo-EM imaging (S1 Fig and S1 Table). Analysis of particle distributions with 3D classification showed that CD163 self-assembles into high-order structures such as dimers, trimers and tetramers. Given that the CD163 trimer was the most prominent species with the most distinguishable features in the dataset, we performed 3D refinement on the trimer, which resulted in a cryo-EM map at a resolution of 3.0 Å (Fig 1A, 1B).

The structural model built from this data shows that each individual subunit of the CD163 trimer adopts a ladle-like shape with a dimension of 80 Å × 65 Å × 175 Å. The first four SRCR domains (D1–D4) extend away from the membrane, forming a handle, while the last five SRCR domains (D5–D9) near the membrane surface create a compact, bowl-like structure (S2A, 2B Fig). The positioning and association angles of these nine domains are remarkably rigid, remaining nearly identical in each subunit. However, the disposition of the subunits in the trimer, designated as CD163-I, CD163-II, and CD163-III, is not symmetrical; instead, they intertwine into a structure resembling a tilted cylinder (Fig 1A, 1B). The trimer is held together by four direct interdomain contacts involving domains D3, D7 and D9. Chains CD163-I and -II pair through D3–D3 and D7–D9 interactions, CD163-II and -III interact through D3–D9 contacts and finally, CD163-III and -I contact each other through D7–D9. All contacts involve electrostatic pairing interactions coordinated by a total of eight $Ca^{+2}$ ions, which are clearly visible in the electron density maps (Figs 1C–1F and S2C). Thus, the calcium binding motifs in domains D3, D7, and D9 (S2D Fig) set the stage for the trimeric structure of unliganded CD163. As elaborated further below, this configuration is not compatible with the CD163-HpHb complex formation or ligand binding and therefore represents an autoinhibitory state.

Given that $Ca^{+2}$-mediated electrostatic interactions are present at each subunit interface, we examined the oligomeric states of the CD163 ECD after removing $Ca^{+2}$ with EDTA. Through size exclusion chromatography coupled with multi-angle light scattering (SEC-MALS), the CD163 ECD was found to have an average mass of 398 kDa, indicating the formation of a CD163 trimer. However, upon addition of EDTA, the average mass dropped to 140 kDa, corresponding to a CD163 monomer (Fig 1G). Furthermore, mass photometry detected dimer, trimer, and tetramer species of the CD163 ECD, although a pronounced peak for monomers was also observed, likely due to sample dilution causing dissociation of CD163 oligomers before the measurement. In contrast, only CD163 monomers were observed when $Ca^{+2}$ was removed (Fig 1H). In summary, findings from both methods confirm that $Ca^{+2}$-dependent electrostatic interactions are essential for CD163 self-assembly.

### Overall architecture of the CD163/HpHb complex

To investigate the CD163/HpHb interaction, we next generated CD163/Hp(1–1)Hb complex by mixing free CD163 ECD with commercially available Hp(1–1) and Hb proteins. The complex, isolated through size-exclusion chromatography, was validated using gel electrophoresis to confirm the presence of all three components (S3A, S3B Fig) and subsequently subjected to cryo-EM image acquisition and data analysis (S4 Fig and S2 Table).

The cryo-EM map of the complex, obtained at a resolution of 3.3 Å, enabled us to construct an atomic model that clearly demonstrates CD163 and the dimeric Hp(1–1)Hb bind with a 3:1 receptor-ligand stoichiometry. It also shows that

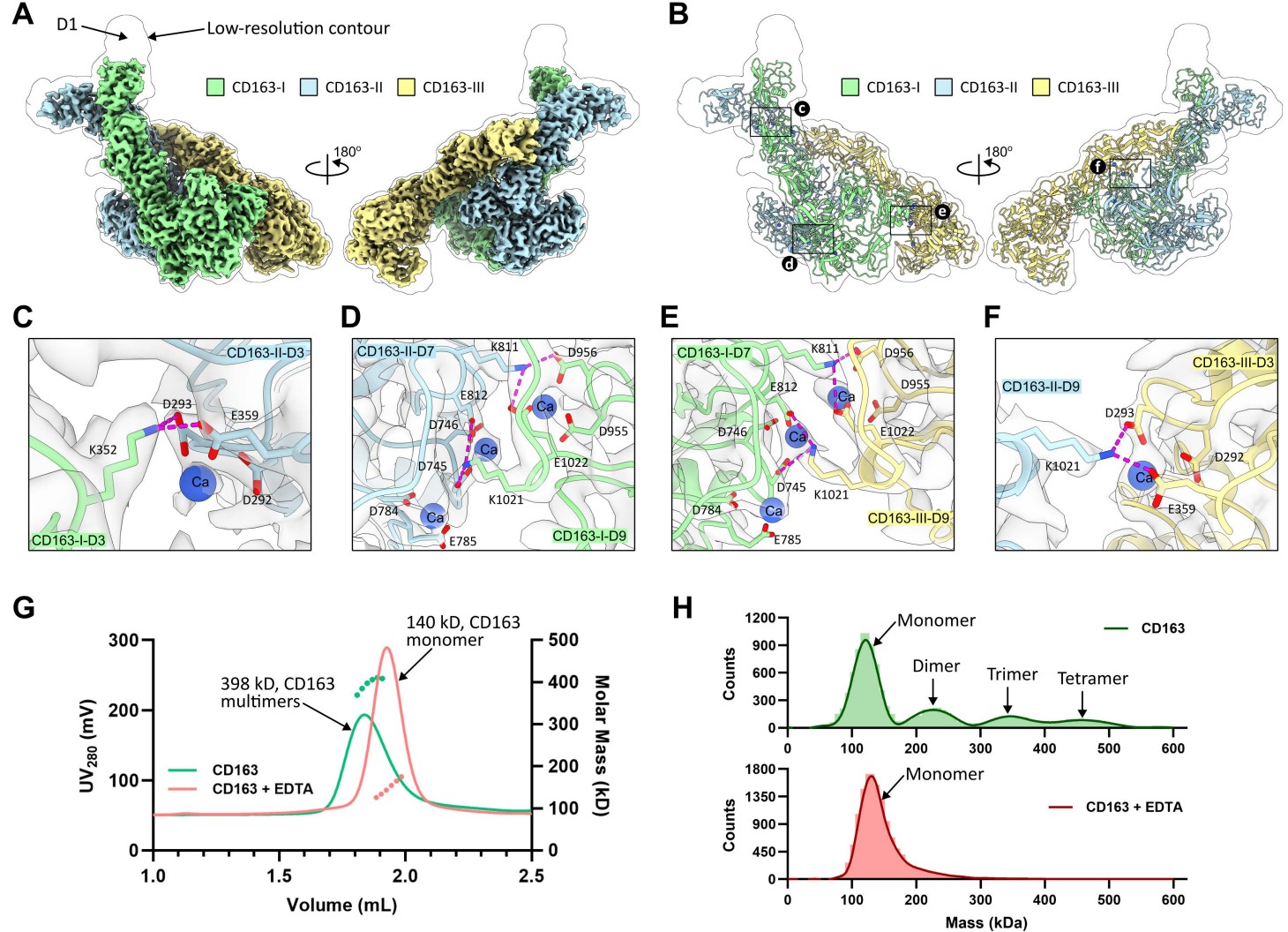

**Fig 1. Cryo-EM structure of the CD163 homotrimer. (A)** Cryo-EM map for CD163 trimer. Two side views, with each subunit in different color, are shown. A map in transparent representation is contoured at a low level to show electron density for the unresolved flexible domain D1. **(B)** Molecular model of the CD163 trimer fitted into the low-contour-level map. **(C–F)** Close-up views of the inter-subunit interactions. The key residues and Ca$^{+2}$ ions involved in these interactions are shown as sticks and spheres, respectively. Magenta dashed lines indicate salt bridges between Lys and Asp/Glu residues. The cryo-EM map in these regions is shown as gray surface. **(G)** SEC-MALS analysis of CD163 ECD with and without adding EDTA. **(H)** Mass photometry analysis of 50 nM of CD163 ECD with and without adding EDTA. Source data for **(G–H)** can be found in S1 Data.

only the Hp SP domain (HpSP) and Hbαβ from one protomer of Hp(1–1)Hb, termed the HpHb triad, interact with the three subunits of CD163 (Fig 2A, 2B). The final model includes all the structural domains that participate in direct contact within the 3:1 complex. The distal regions such as domains D1 of each CD163 subunit, D2 of CD163-III, and the second protomer of Hp(1–1)Hb which are not part of the ligand binding interface, are poorly resolved in the structure due to insufficient density, likely caused by intrinsic dynamics (Fig 2B).

In contrast to the asymmetrical assembly observed in the unliganded CD163 trimer, the three subunits of CD163 in the CD163/Hp(1–1)Hb structure arrange in a nearly 3-fold symmetry, resembling a triangular prism with an overall dimension of 160 Å × 220 Å × 140 Å (Fig 2A, 2B). Here, D5–D9 from each CD163 subunit join together to form an enclosed base

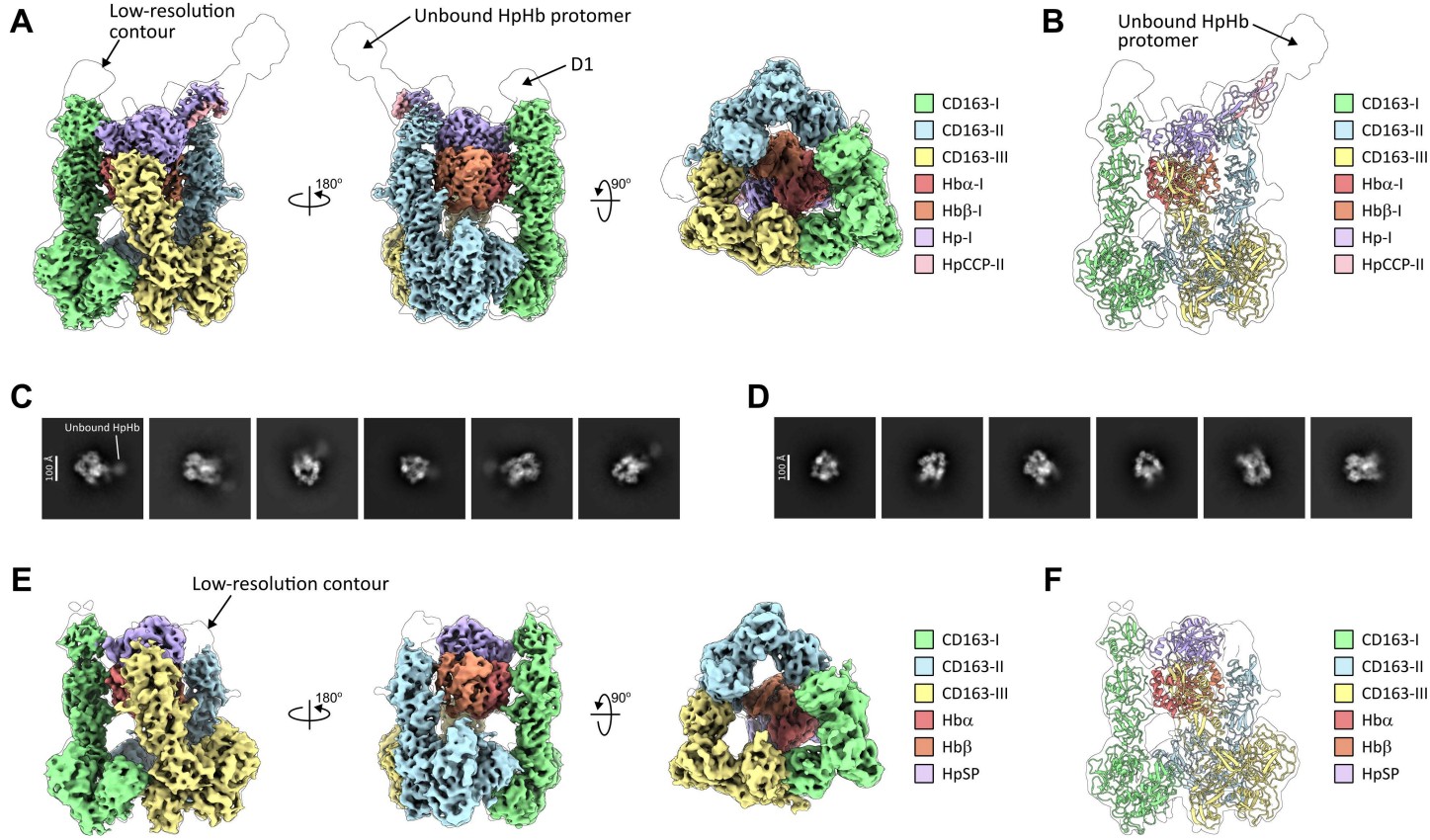

**Fig 2. Cryo-EM structures of the CD163/Hp(1–1)Hb and CD163/HpSPHb complexes. (A)** Cryo-EM map of the CD163/Hp(1–1)Hb complex colored by each component and shown in two side views and one bottom view. A map in transparent representation is contoured at a low level to show electron density for the unresolved flexible domain D1 and unbound HpHb protomer. **(B)** Molecular model of the CD163/Hp(1–1)Hb complex fitted into the low-contour-level map. **(C–D)** Representative 2D class averages of CD163/Hp(1–1)Hb **(C)** and CD163/HpSPHb **(D)**. The protruding unbound HpHb protomer is seen in the 2D class averages of CD163/Hp(1–1)Hb but not in the 2D class averages of CD163/HpSPHb. **(E)** Cryo-EM map of the CD163/HpSPHb complex colored by each component and shown in two side views and one bottom view. A map in transparent representation is contoured at a low level to show electron density for the flexible peripheral regions. **(F)** Molecular model of the CD163/HpSPHb complex fitted into the low-contour-level map.

near the membrane, stabilized by interdomain contacts between neighboring D7 and D9, while D1–D4 of each CD163 extend upward to embrace the HpHb triad. This interaction positions the HpSP domain at the apex of the complex, with Hbαβ nestled below between D3 and D4 of each CD163. As a result, the HpHb triad presents distinct surfaces at each symmetrical contact point with CD163 (Fig 3A).

The fact that the HpHb complex presents only one of its protomers for interaction with a CD163 trimer, prompted us to examine whether a deletion mutant HpSPHb lacking the CCP domain and containing just one Hbαβ bound to the truncated HpSP domain would bind in a similar fashion as the dimeric Hp(1–1)Hb protein. To test this, we generated the HpSPHb protein, prepared the CD163/HpSPHb complex (S3C, S3D Fig), and performed cryo-EM analysis of this complex (S5 Fig and S2 Table). Initial 2D class analysis showed that the CD163/HpSPHb complex had 2D class averages similar to those of the CD163/Hp(1–1)Hb complex, but without protrusion features that could account for a second HpHb protomer (Fig 2C, 2D). A map obtained at 4.0 Å resolution was used to build the structure of the CD163/HpSPHb complex encompassing domains D2–D9 of CD163-I, D3–D9 of CD163-II and CD163-III, and one bound HpSPHb (Fig 2E, 2F). This structure, when compared to the CD163/Hp(1–1)Hb complex, clearly shows that HpSPHb has the same 3:1 binding mode

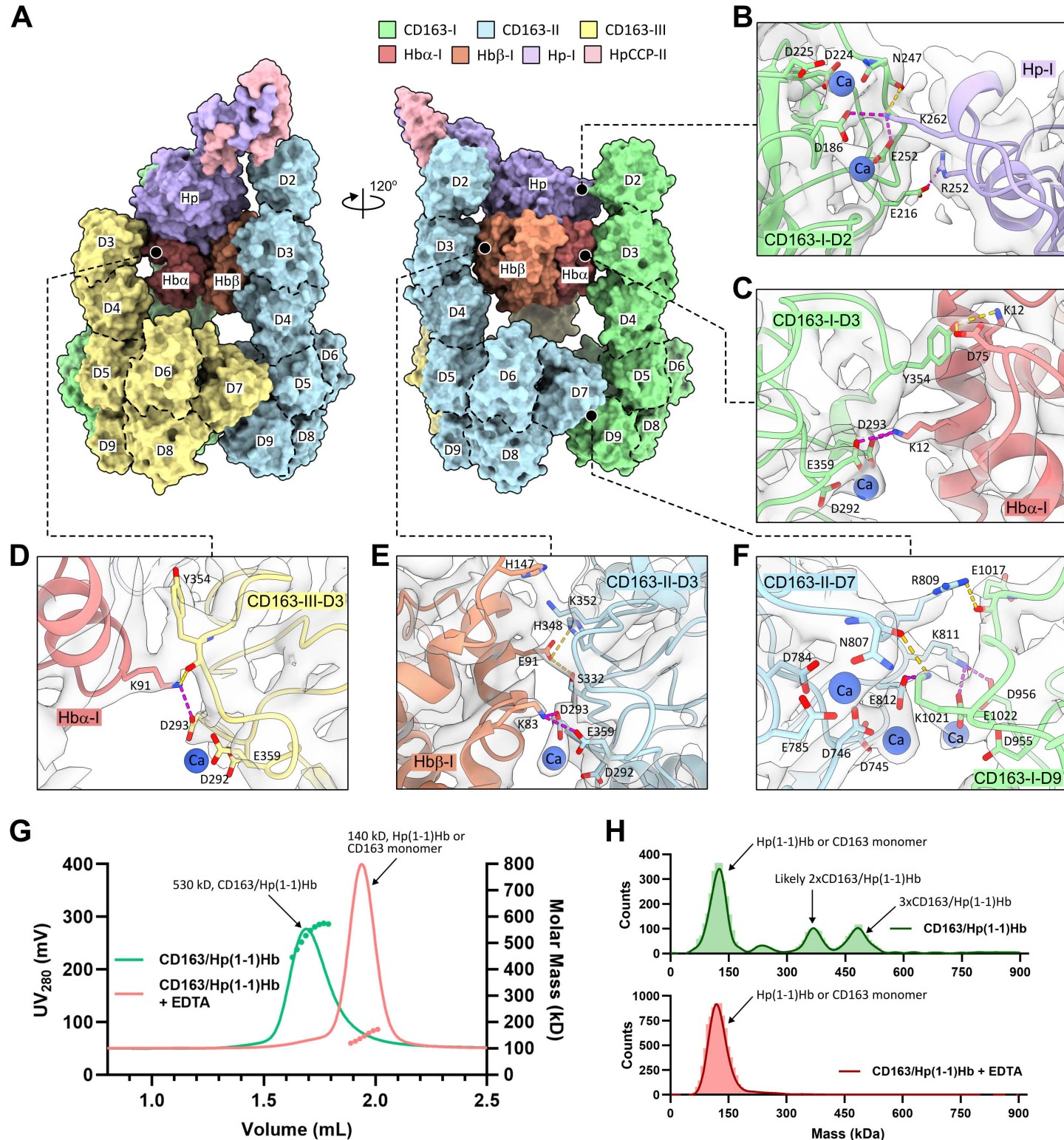

Fig 3. **Ubiquitous Ca+2-mediated electrostatic interactions in the structure of the CD163/Hp(1–1)Hb complex. (A)** Surface representation of the CD163/Hp(1–1)Hb structure colored by each component. CD163 SRCR domains (D2–D9) are labeled on the surface. **(B–F)** Close-up views of the interactions between the CD163 trimer and HpHb protomer **(B–E)** and the interactions between CD163-I D9 and CD163-II D7 **(F)**. The key residues and Ca+2

ions involved in these interactions are shown as sticks and spheres, respectively. Magenta dashed lines indicate salt bridges between Lys/Arg and Asp/Glu residues. Yellow dashed lines indicate hydrogen bonds. The cryo-EM map in these regions is shown as gray surface. **(G)** SEC-MALS analysis of CD163/Hp(1–1)Hb with and without adding EDTA. **(H)** Mass photometry analysis of 50 nM of CD163/Hp(1–1)Hb with and without adding EDTA. Source data for **(G–H)** can be found in S1 Data.

and identical binding interactions with the CD163 trimer as Hp(1–1)Hb (Fig 2A, 2B, 2E, 2F). These observations support the notion that only one protomeric unit of Hp(1–1)Hb is required for engagement with CD163 trimer.

## Ca$^{+2}$-dependent electrostatic network in the CD163/HpHb complex

The CD163/Hp(1–1)Hb complex features a large composite interface made up of two types of binding interactions: those that facilitate ligand recognition and those that drive the self-association of receptor subunits. As expected from previously published data [17,18] and in accordance with our findings reported here, both types of interactions heavily rely on a common pattern of calcium-dependent electrostatic interactions (Fig 3A–3E).

The ligand recognition interface involves contacts with each subunit of the HpHb triad. At the top of the interface, the HpSP domain contacts D2 of CD163-I. Here, the positively charged Lys262 from the edge of Hp forms electrostatic pairing with the negatively charged Ca$^{+2}$-coordinated acidic cluster Asp186 and Glu252, while Arg252 nearby makes a salt bridge interaction with Glu216 (Fig 3B). These observed interactions corroborate the results from previously published mutagenesis data, showing that Lys262 and Arg252 of Hp and the acidic cluster residues from CD163 D2 and D3 are essential for the interaction between CD163 and HpHb [18]. Directly below HpSP, the Hbα subunit forms contacts on its opposite sides, bridging the D3 domains from the two CD163 chains, CD163-I and CD163-III, together. On one side, the Ca$^{+2}$-coordinated acidic cluster Asp293 and Glu359 from D3 of CD163-I makes electrostatic pairing with Lys12 of Hbα, and on the opposite side, the same Ca$^{+2}$-coordinated cluster from D3 of C163-III makes electrostatic paring with Lys91 of Hbα (Fig 3C, 3D). Lastly, Hbβ engages with D3 of the remaining third chain, CD163-II, through electrostatic pairing of Lys83 with the Ca$^{+2}$-coordianted acidic cluster Asp293 and Glu359 of CD163-II (Fig 3E). Importantly, Hbα and Hbβ together make up approximately two-thirds of the contact surface with CD163, with the remaining one-third contributed by HpSP. This might explain why Hb can bind CD163 and be internalized for degradation even in the absence of Hp [20].

The CD163–CD163 interface at the bottom follows the same pattern of Ca$^{+2}$-dependent electrostatic interactions that recur between D7 and D9 of adjacent subunits. At each interface, there are three Ca$^{+2}$ ions surrounded by three clusters of charge-complementary interactions. D7 contributes four residues: Lys812, Arg1021, and Asp746, Glu812, which form reciprocal hydrogen bonds and salt bridges with four residues from D9: Asp956, Glu1022, Glu1017, and Lys1021, respectively (Figs 3F and S6A). To understand if the self-assembling interactions of CD163 enhance binding to HpHb, we compared the binding affinities of Hp(1–1)Hb to the truncated CD163 protein (D1–D5) and the full length CD163 ECD. Surface plasmon resonance (SPR) measurements demonstrated that the full-length CD163 ECD had about 10-fold stronger binding affinity compared to the truncated CD163 protein lacking the self-associating domains (S6B–S6D Fig).

Considering the central role of structural Ca$^{+2}$ ions in both the ligand-binding interface and the self-associating receptor interface, we assessed the stability of the CD163/Hp(1–1)Hb complex upon removing Ca$^{+2}$ ions. According to SEC-MALS measurements, the mass of the CD163/Hp(1–1)Hb complex is approximately 530 kDa, which corresponds to one Hp(1–1)Hb dimer bound with three CD163 molecules (Fig 3G). After removing Ca$^{+2}$ ions, the mass dropped to 140 kDa, indicating the complex collapsed into individual Hp(1–1)Hb or CD163 components. Likewise, mass photometry of the diluted CD163/Hp(1–1)Hb sample showed peaks corresponding to an Hp(1–1)Hb dimer complexed with either two or three CD163 molecules, as well as a peak representing an individual Hp(1–1)Hb or CD163 molecule (Fig 3H). However, upon Ca$^{+2}$ ion removal, only the peak for an individual Hp(1–1)Hb or CD163 molecule remained. These results confirm that the extensive electrostatic interactions observed in the structure heavily rely on Ca$^{+2}$ ions, underscoring their vital role in CD163/HpHb assembly.

## Structural changes accompanying HpHb binding

Comparative analysis of the CD163 trimers in the unliganded and Hp(1–1)Hb-bound structures clearly demonstrates that in the absence of ligand, the upper ligand binding arms of CD163 swing into the vacant ligand space, engaging in auto-inhibitory interactions, thus creating both direct and steric obstruction of the ligand binding site (Figs 1A, 1B and 2A, 2B). Therefore, a conformational change is required to allow ligand entry. Indeed, the three subunits in the CD163/Hp(1–1) Hb complex show a significant shift in their relative positions, primarily through concerted swing motions at the base (S1 Movie and Fig 4A–4D). By making this adjustment, the ligand-binding arms assume their primed conformation in the 'up' position to wrap around the ligand and simultaneously seal the base to the symmetrical structure at the bottom (Fig 4E). The swing-switch transition between inactive and active configurations can be deconvoluted in two main processes: (1) large swing motions of adjacent subunits along domains D5–D9 facilitating the opening and closing of the base (Fig 4B–4D) and (2) additional small bending of the ligand binding arms allowing them to close and open (Fig 4B). Considering that all these motions revolve around the calcium ion binding motifs, recapitulating the same patterns of the $Ca^{+2}$-mediated interaction (Figs 1D, 1E, 3F, and S6A), the calcium-mediated network appears to be key to structural flexibility during the transition between the two states.

## Mechanism of HpHb scavenging by CD163

Based on this study and previous biochemical data, we propose the following possible sequence of structural events that result in clearance of HpHb by CD163 (Fig 5). At higher local concentrations on the cell surface, CD163 forms predominantly an autoinhibitory trimer prior to ligand binding. This pre-ligand "inactive" assembly could constrain the conformational dynamics of CD163 to prevent the uptake of random weak ligands with non-specific binding properties. During the uptake of free HpHb, CD163 trimers switch from mainly inactive to active state contacts, employing a uniform recognition mechanism across different multimeric forms of HpHb. The key aspect of ligand recognition in this mechanism is that each CD163 trimer presents a composite but singular binding site dedicated to only one protomeric unit of HpHb. Such an uptake pattern seems highly effective as it ensures uniform transport and equal endocytosis of all HpHb isoforms.

Given that calcium is required for the formation of both ligand-free and ligand-bound CD163 assemblies, a differential calcium ion concentration across cellular compartments is essential for cargo release. Compared to extracellular spaces, the endosomal lumen has mildly acidic pH and substantially lower calcium concentration conditions [21]. Therefore, throughout the ongoing constitutive or ligand-dependent receptor internalization and recycling processes between the plasma membrane and early endosomes [12], the CD163 trimers loaded with HpHb complexes disassemble into individual subunits upon reaching the endosome, resulting in subsequent ligand release. Released HpHb are subsequently trafficked to lysosomes for degradation, whereas CD163 recycles back to the cell surface for further rounds of HpHb uptake and clearance.

## Discussion

While interactions of HpHb with CD163 have been extensively investigated [16–18], the cryo-EM structures described here provide a complete view of the extracellular assembly between CD163 and HpHb. The overall architecture of the 3:1 complex arranges the three CD163 molecules around one dimeric HpHb molecule in such a way that only one protomer of HpHb is sufficient for ligand-recognition and subsequent CD163-mediated uptake (Fig 2). This mechanism should be applicable to other multivalent forms of HpHb, irrespective of the binding valency in the higher-order HpHb formats. Consequently, through this uniform uptake, CD163 can enable highly efficient transport and equal endocytosis of all HpHb multimers.

The structure demonstrates that the ligand-binding arms of CD163 are positioned to coordinate the Hp(1–1)Hb complex through a composite binding site, dictated by multiple relatively weak interfaces from each binding component (Fig 3A–3E). This implies its highly promiscuous nature, able to accept and adapt to different ligands. Such adaptability is

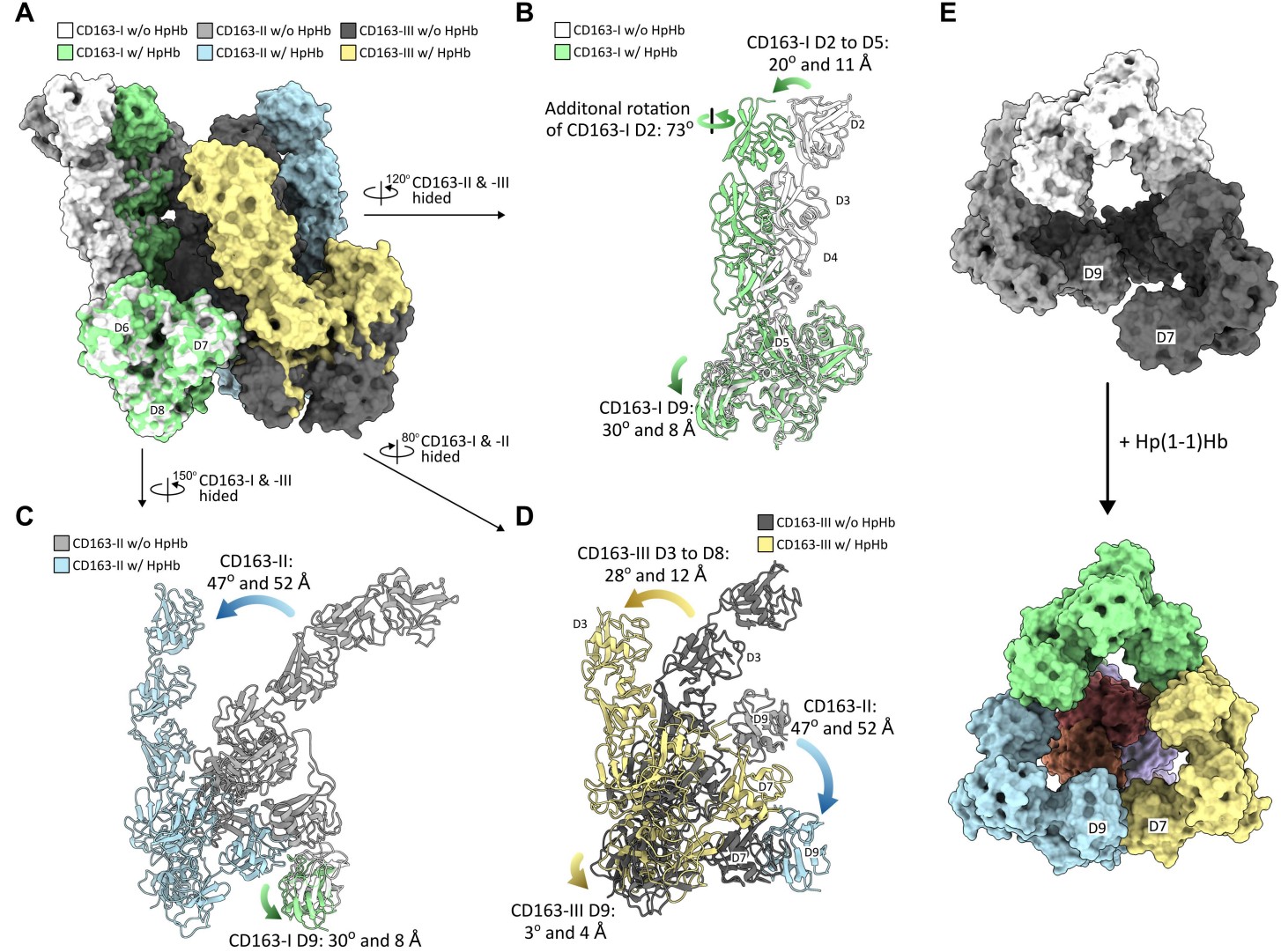

**Fig 4. Structural comparison between the unliganded and Hp(1–1)Hb bound CD163 trimer. (A)** Structure Superposition of the unliganded and Hp(1–1)Hb bound CD163 trimer based on the alignment of D6 to D8 from CD163-I (RMSD = 0.81 Å) to inspect structural motions of CD163 subunits for Hp(1–1)/Hb binding. **(B)** Focused view on CD163-I showing that its D2 to D5 rotates 20° and shifts 11 Å with additional horizontal 73° rotation of the D2, and its D9 rotates 30° and shifts 8 Å. **(C)** Focused view on CD163-II showing that it rotates 47° and shifts 52 Å, accompanying the movement of CD163-I D9 and preserving the Ca$^{+2}$-medicated D7–D9 interactions between CD163-I and CD163-II. **(D)** Focused view on CD163-III showing that its D3 to D8 rotates 28° and shifts 12 Å to establish the Ca$^{+2}$-medicated D7–D9 interactions between CD163-II and CD163-III. At the same time, CD163-III D9 slightly rotates 3° and shits 4 Å, preserving the Ca$^{+2}$-medicated D7–D9 interactions between CD163-I and CD163-III. **(E)** bottom view of the CD163 trimers showing that D9 of CD163-II connects with D7 of CD163-III to form an enclosed structure upon Hp(1–1)Hb binding. Each CD163 subunit is colored as in (A–D). Hp(1–1)Hb is shown as surface and colored as in Fig 3.

particularly significant given that CD163 is a multifunctional receptor that binds and responds to a wide variety of different ligands [22]. Therefore, it can be argued that the trimeric assembly of CD163, as observed in the CD163/Hp(1–1)Hb complex, may serve as a fundamental biological unit in generating different activation outputs upon interacting with distinct ligands. In this regard, the reported interaction of CD163 with the soluble form of tumor necrosis factor-like weak inducer (TWEAK) is noteworthy [23]. TWEAK is a classical TNF family trimeric cytokine, to which CD163 binds to, internalizes,

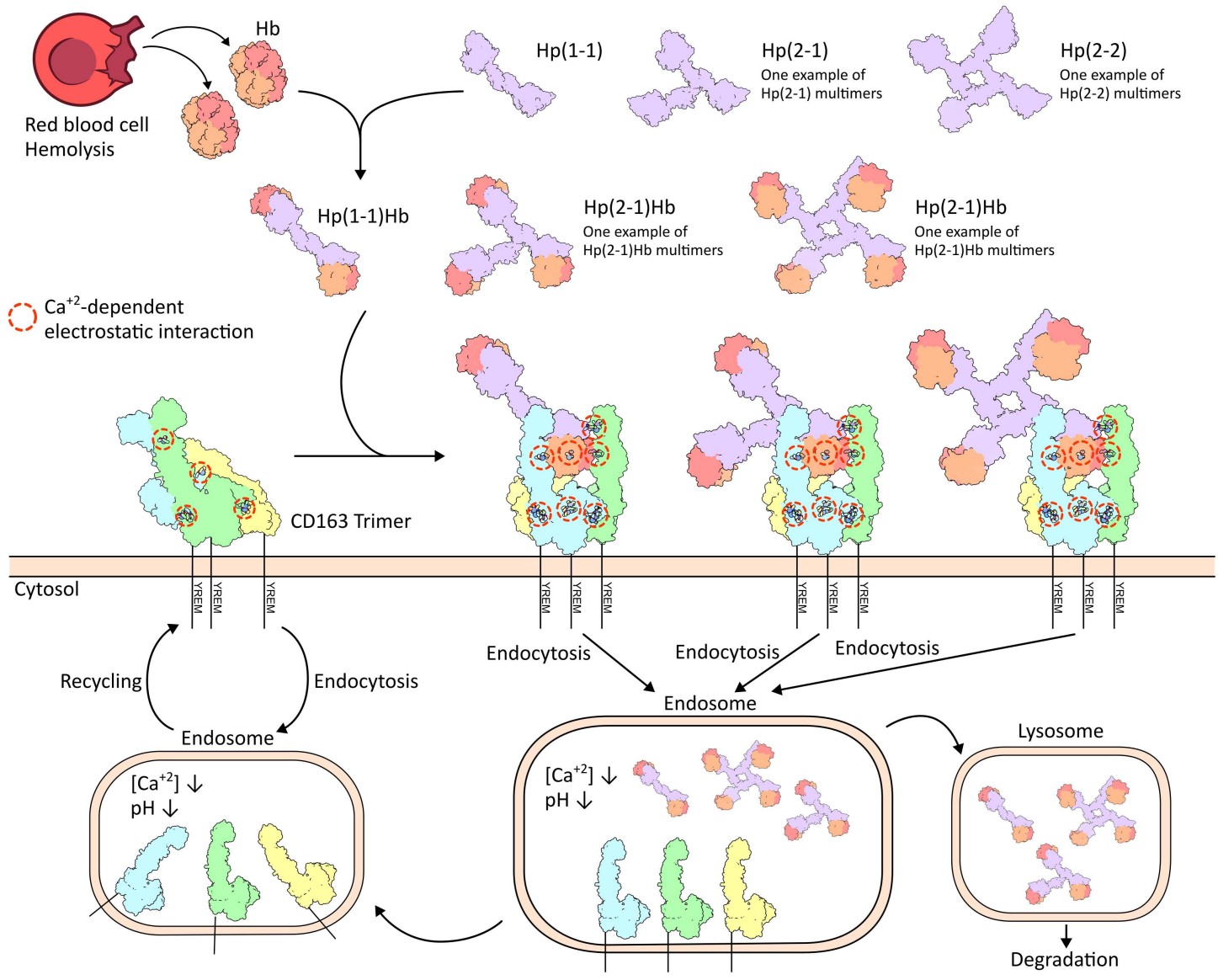

**Fig 5. Schematics of proposed mechanism for haptoglobin–hemoglobin clearance mediated by CD163.** During intravascular hemolysis, haptoglobins seize released hemoglobin forming HpHb complexes. Only dimeric Hp(1–1)Hb, trimeric Hp(2–1)Hb and tetrameric Hp(2–2)Hb structures are shown here for simplicity. On the cell surface, CD163 predominantly forms autoinhibitory trimers prior to ligand binding. During HpHb uptake, CD163 transitions from inactive to active state trimers through a uniform recognition mechanism across different multimeric forms of HpHb. During the ongoing constitutive (left) or ligand-dependent (right) receptor internalization and recycling, the CD163-HpHb complexes disassemble upon reaching the endosome (due to low pH and low $Ca^{+2}$ concentration), resulting in subsequent ligand release. The dissociated HpHb complexes are transferred to lysosome for degradation, while the dissociated receptors are recycled back to the cell surface and reassemble into CD163 trimers.

and clears from circulation [24]. Our quick analysis indicates that the spatial configuration of the CD163 ligand binding site matches the overall geometry and dimensions of the trimeric TWEAK structure, suggesting that CD163 could engage TWEAK in a 3:3 stoichiometry. However, further additional studies are required to confirm or refute this speculation.

The formation of the enclosed trimeric base near the membrane is also of particular interest. The CD163/Hp(1–1)Hb structure combined with AlphaFold structure prediction modeling [25] suggests that this triangular shape structure can

provide a spatial platform for downstream internalization as well as signaling events by enabling the three copies of the transmembrane helical domain to converge (S7A Fig). In this model, each membrane-proximal D9 domain is connected to the transmembrane helix via a 15-residue long linker. At this length, each of the three linkers can cover a distance of roughly 40 Å, allowing them to converge at the center and promote the formation of a transmembrane helical trimer. The observed geometry of the CD163 extracellular trimer is therefore poised to induce the proximity of the cytoplasmic tails in a way that can promote endocytosis and other downstream processes. This model also implies that trimerization of the transmembrane domains could reinforce the formation of CD163 trimers on the cell surface, which may be one of the factors why soluble shed variants of CD163 are not effective at outcompeting the HpHb uptake by the membrane-bound CD163 [26].

In light of the above discussion, we next questioned whether a single higher-order HpHb multimer could be simultaneously captured by two or more receptor trimers on the cell surface. Our structural analysis indicates that such capturing is not feasible for the dimeric Hp(1–1)Hb due to the binding geometry of the CD163/Hp(1–1)Hb complex, in which the second free HpHb protomer extends up and away from the membrane (Figs 2A, 2B and S7A). We then utilized AlphaFold structure prediction tools [25] to generate models for the structures of HpHb that contain isoform Hp2, specifically the higher multimeric Hp(2–1)Hb and Hp(2–2)Hb complexes. Follow-up structural docking analysis revealed that in the case of Hp(2–1)Hb and Hp(2–2)Hb, two protomers in each multimeric mixture can project their HpHb triads toward the membrane, allowing simultaneous engagement with two CD163 trimers on the cell surface (S7B Fig). According to this model, at least two HpHb triads in a higher multimeric form of HpHb are captured by two CD163 trimers, creating an avidity effect. This aligns well with studies demonstrating that CD163 binds more strongly to multivalent Hp(2–1)Hb and Hp(2–2)Hb than to dimeric Hp(1–1)Hb [2]. Along these lines, previous studies on cell-surface CD163 cross-linking have shown that receptor signaling and activation are triggered when cross-linked by agonistic antibodies [13,27]. It is plausible that oligomeric assemblies beyond trimers are favored in the native membrane environment when crosslinked by a multivalent ligand, as modeled here, leading to higher-order receptor clustering on the cell surface. Such multivalent clustering is likely to increase the proportion of CD163/HpHb complexes ready to be internalized, which may explain why Hp(1–1)Hb complexes are scavenged more rapidly than Hp2 containing Hp(2–1)Hb and Hp(2–2)Hb complexes [28]. Other differences that have been reported between Hp(1–1)Hb and higher multimeric HpHb forms relate to their distinct effects on macrophage activation, resulting in expression of proinflammatory or anti-inflammatory cytokines [28]. Whether higher-order versus low-order CD163-HpHb clusters can elicit different signaling responses while destined for endocytosis is currently unknown and may deserve further investigation. The general advantages of receptor clustering have been noted for CD36, a scavenger receptor involved in LDL uptake by macrophages [29], and are widely reported in the activation mechanisms of various immunoreceptors [30]. However, more comprehensive and detailed investigation is required to understand the events mediated by CD163 after ligand binding and their impact on the outcome of ligand-CD163 interactions.

Another interesting aspect revealed by our studies is that CD163 can form self-associating trimers prior to ligand binding. The three subunits of CD163 associate differently in pre-ligand state, with the ligand binding arms docked against each other in an autoinhibitory asymmetric configuration. This points to a receptor masking mechanism that might be employed to minimize or prevent non-specific binding of CD163 to random ligands. The swing-switch transition between inactive and active configurations can be summarized as a concerted motion between two swapped states: 'closed-arms + open-base' and 'open-arms + closed-base', respectively.

Our findings also clarified the exceptional function of calcium ion coordination in mediating both pre-ligand and ligand-receptor complex assemblies. We identified a total of eight and 14 structural $Ca^{2+}$ ions involved in direct interface contacts in the ligand-free and ligand-bound structures, respectively. Furthermore, our biochemical assays demonstrated the crucial role of calcium ions in stabilizing the CD163/Hp(1–1)Hb complex, as the addition of calcium-chelating EDTA caused the collapse of the complex into individual components. Taken together, this indicates that CD163 is competent of ligand binding only when it is exposed to calcium at physiological pH, as is the case when it resides on the cell surface.

Finally, the prevalent use of calcium-ions in key interfaces may be a critical factor contributing to the structural plasticity and adaptability at these interfaces to support the CD163 function dynamics such as ligand binding and ligand release.

Lastly, there is evidence from our biophysical assays for monomeric, dimeric and tetrameric species of CD163-ECD in solution. We also observed cryo-EM 3D classes corresponding to dimeric and tetrameric configurations of CD163 in the unliganded state (S1A Fig) and dimeric configuration of CD163 in the Hp(1–1)Hb-bound state (S4A Fig). However, the electron density maps derived from the dimeric classes exhibited weak densities for the third CD163-III chain, observed only at low contour levels, in both structures. The two remaining CD163-I and CD163-II chains, with strong densities, displayed the same self-inhibitory and ligand-binding interactions as in the two respective trimeric structures. These observations led us to conclude that the dimeric assemblies of unliganded and ligand-bound CD163 are essentially the same as the trimeric structures, but with reduced occupancies for the third chain. As for the tetrameric species, we argue that the detected species are likely artifacts of utilizing truncated CD163-ECD. The linker between CD163-ECD and the TM domain is not long enough to allow for the tetramerization of the membrane-tethered form of CD163 as observed in the 3D classes. Therefore, we believe that CD163 requires trimerization for function, with the trimer serving as the primary structural unit for its scavenging role.

While our work was in preparation for submission, two independent groups reported the cryo-EM structures of the CD163/Hp(1–1)Hb complex [31,32]. One of these studies also described the structures of unliganded forms of CD163 [32]. Our work, while generally aligning with the findings from the above reports, extends them first by examining these structures at improved local resolution, achieved through sophisticated focused 3D classifications and local refinements (S4 Fig). This greater structural detail enabled us to unambiguously assign the electron density, specifically at contact interfaces, to all critical $Ca^{+2}$ ions and all $Ca^{+2}$ coordinating side chain interactions. Second, we have constructed and proposed structural models on which we base our discussion around receptor clustering when bound to higher multimeric forms of HpHb.

## Materials and methods

### CD163 Protein expression and purification

DNA encoding the extracellular domain (residues 42–1,045) of Human CD163 (Uniprot Q86VB7) with a N-terminal murine HC signal peptide (MGWSCIILFLVATATGVHS) and a C-terminal 8× His tag was cloned into a pTT5 expression vector. The vector was transfected into Expi293 cells (Thermo Fisher Scientific) at 37°C for transient expression according to the manufacturer's instructions (Thermo Fisher Scientific). The cells were maintained shaking at 30°C after 18 h of transfection and the cell medium was harvested for purification after 96 h of transfection. CD163 protein was purified with nickel-charged IMAC resins (Cytiva) and eluted in IMAC buffer (50 mM Tris, pH 7.5, 500 mM NaCl, 5 mM $CaCl_2$, 10% glycerol, 250 mM Imidazole, 50 mM Arg, and 50 mM Glu). The protein was then polished by size-exclusion chromatography (SEC) on a HiLoad 16/600 Superdex 200 pg (Cytiva) in SEC buffer (20 mM HEPES, pH 7.0, 300 mM NaCl, 5 mM $CaCl_2$, 50 mM Arg, and 50 mM Glu). For cryo-EM studies, the CD163 protein was further exchanged into EM buffer (20 mM HEPES, 150 mM NaCl, pH 7.5, and 2 mM $CaCl_2$.) and concentrated to 4 mg/ml.

### HpSP expression and purification

The gene encoding the serine protease (SP) domain (residues 147–406) of Human Haptoglobin (Uniprot P00738) with a N-terminal GP47 signal peptide (MLLVNQSHQGFNKEHTSKMVSAIVLYVLLAAAAHSAFA) and 6× His tag, was subcloned into a pFastBac1 vector for baculovirus-driven expression in Sf9 insect cells (Thermo Fisher Scientific). Bacmid and baculovirus were generated with the pFastBac1 construct and Sf9 cells were infected at 27°C using standard protocols. The cell medium was harvested after 48–55 h of infection and passed through a 0.2 μm filter. HpSP protein was purified with a HisTrap Excel column (Cytiva) and eluted in phosphate-buffered saline pH 7.2 with 250 mM Imidazole. The protein was further polished by SEC using a HiLoad 16/600 Superdex 200 pg column (Cytiva) equilibrated with 20 mM HEPES pH 7.0 and 150 mM NaCl.

## Complex formation and purification

Haptoglobin Phenotype 1-1 [Hp(1–1)] and Hemoglobin (Hb) were purchased from Sigma Aldrich. To generate CD163/Hp(1–1)Hb complexes, Hp(1–1), Hb and CD163 were mixed at the molar ratio of 1:2:3, which showed little excess of un-complexed material, and incubated overnight at 4°C. The complex was purified by SEC using a HiLoad 16/600 Superdex 200 pg column (Cytiva) equilibrated with EM buffer. The purity of the purified complex was determined by SDS-PAGE. The peak fraction from the SEC purification containing 0.2 mg/mL of the CD163/Hp(1–1)Hb complex was used for cryo-EM sample preparation.

The CD163/HpSPHb complex was generated and purified in a similar manner as above. The peak from the SEC purification containing 0.47 mg/mL of the CD163/HpSPHb complex was used for cryo-EM sample preparation.

## SEC-MALS analysis

Purified proteins or complexes were injected onto a Superose 6 Increase 5/150 GL column (Cytiva). Experiments were performed on a HPLC 1,260 Infinity system (Agilent) connected in series with a Wyatt miniDAWN TREOS 3 angle-static light-scattering detector, and a Wyatt Optilab rEX refractive index detector at flow rate 0.35 mL/min. Data were analyzed with ASTRA 6.1 software (Wyatt Technology). The dn/dc (refractive index increment) value for all samples was defined as 0.185 mL/g, which is a standard value for proteins.

## Mass photometry

Mass photometry measurements were conducted using a Refeyn OneMP mass photometer (Refeyn Ltd, UK). Glass coverslips (24 × 50 mm, No. 1.5H; Marienfeld GmbH and Co. KG, Germany) were cleaned sequentially with Milli-Q water, isopropanol, and a second rinse with Milli-Q water. Coverslips were then dried using a stream of filtered air to remove any residual contaminants. To create sample wells, CultureWell gaskets (3 mm diameter × 1 mm depth; Grace Bio-Labs, USA) were affixed to the cleaned coverslips.

The buffer solutions used for the experiments included 20 mM HEPES pH 7.5, 100 mM NaCl, and 2 mM $CaCl_2$, with or without 2 mM EDTA, depending on the experimental conditions. For blank measurements, 16 μL of the buffer was added to a single well prepared on the coverslip. Frozen samples of CD163 and CD163/Hp(1–1)Hb were thawed immediately prior to measurement. The samples were diluted in imaging buffer and subsequently mixed with the buffer already present in the well, resulting in a final concentration of 6–12 nM and a total volume of 20 μL per well.

Measurements were recorded for 1 min using the AcquireMP software (Refeyn, UK). Data analysis was performed with DiscoverMP software (Refeyn, UK) to determine mass distributions. Each sample was measured independently in triplicate ($n = 3$) to ensure reproducibility.

## Affinity determination for CD163 binding haptoglobin–hemoglobin by SPR

The Assay was performed on a Biacore 8k system (Cytiva). Haptoglobin chip was prepared by capturing biotinylated Hp(1–1) to a Biacore SA series S sensor chip that was preconditioned by three 60-s injections of 50 mM NaOH and 1 M NaCl at 10 mL/min. The finished chip had 8 flow cells with haptoglobin immobilized. Kinetic assays were performed in running buffer HBST+ supplemented with 1 mg/mL BSA and 5 mM $CaCl_2$ at 37°C. Hemoglobin (Sigma) were captured for 2 min at 200 nM diluted into running buffer onto an a haptoglobin surface with a flow rate of 10 mL/min. Three start-up cycles were run first to equilibrate the system. CD163 was injected for 2 min at 30 mL/min and dissociation was monitored for 10 min. Injected samples were diluted into running buffer and a serial dilution series was created with a 3-fold dilution factor to give concentrations of 0, 25, 74, 222, 666 and 2,000 nM. Buffer or the second highest concentration of CD163 was run in duplicate and all interactions were investigated in triplicates. The SA chip was regenerated with three 60-s injections of 50 mM EDTA at 10 mL/min in between each analyte binding cycle.

## Cryo-EM sample preparation and data collection

Samples of CD163, CD163/Hp(1–1)Hb, or CD163/HpSPHb were first diluted to either 0.1 or 0.05 mg/mL using EM buffer. Cryo-EM grids were prepared by glow-discharging copper Quantifoil R1.2/1.3,300 mesh grids with a PELCO easyGlow device (Ted Pella) at 12 mA for 45 s. A volume of 3 μL of each sample was applied to the glow-discharged grid and plunge-frozen into liquid ethane using an FEI Vitrobot Mark IV (Thermo Fisher Scientific) with blot force set to 5, 10 s of blotting time, at 4°C and 100% relative humidity. Micrograph movies were collected on a Titan Krios Microscope (Thermo Fisher Scientific) operating at 300 kV equipped with a Falcon 4i direct electron detector operating in counting mode and a Selectris energy filter with a 10 eV slit width. All cryo-EM data in this study were collected using EPU v3.15 (Thermo Fisher Scientific). Data acquisition parameters can be found in S1 and S2 Tables.

## Image processing

Cryo-EM image analysis was performed with the combination of using RELION-5.0 [33] and cryoSPARC v4.4.1 [34]. For CD163, CD163/Hp(1–1)Hb, and CD163/HpSPHb, 20,360, 22,820, and 15,248 movies were collected, respectively. Drift correction and dose-weighting were carried out using RELION's own implementation of MOTIONCOR2 [35]. CTF estimation of motion-corrected micrographs was carried out using CTFFIND4 [36] or patch CTF estimation in cryoSPARC. Particles were picked using Topaz [37], followed by several rounds of 2D classification to remove undesirable particles.

For the CD163 dataset, 1,456,790 particles selected from the 2D classification were used to generate an ab initio model and subjected to 3D classification. Multiple oligomeric states of CD163 were observed during 3D classification. After two rounds of 3D classification, we concluded that CD163 trimers are the most populated and viable species to achieve a high-resolution 3D map. The best class of the CD163 trimer containing 391,535 particles was refined with homogeneous and non-uniform refinement, yielding a consensus map with a nominal resolution of 3.0 Å (Map A). To improve the local density quality in the peripheral regions of the CD163 trimer, we applied masks for local refinement with particle subtraction on the D2–D3 domains of each CD163 subunit and the D6–D8 domains of the third CD163 subunit. This resulted in four local maps with resolutions ranging from 3.1 to 3.5 Å (Maps B to E). Maps A to E were combined into a composite map (Map F) for model building.

For the CD163/Hp(1–1)Hb dataset, 1,905,994 particles selected from 2D classification were used to create three initial models, while 803,114 particles excluded from 2D classification were utilized to produce three decoy models for supervised heterogeneous refinement in cryoSPARC. Subsequently, 1,175,541 particles were selected for another round of heterogeneous refinement using three initial models and a decoy model. After the second round of heterogenous refinement, homogeneous refinement on a subset of 442,667 particles, showing Hp(1–1)Hb in complex with three CD163 subunits, yielded a 3D reconstruction of 3.7 Å resolution. 3D Variability analysis of the subset revealed substantial motions of the complex, particularly on the third CD163 subunit, which affected the quality of the map. To enhance map clarity, the particles were exported to RELION and subjected to focused 3D classification without alignment with a mask around the third CD163 subunit or the rest of the complex. Particles corresponding to the best class covering the rest of the complex were re-imported to cryoSPARC and refined with homogeneous and non-uniform refinement, yielding a consensus map of the complex with a nominal resolution of 3.3 Å (Map G). To further improve the map quality in the CD163/Hp(1–1)Hb interaction interfaces, local refinement with particle subtraction on the D2–D3 of first CD163 subunit and the D2–D3 of the second CD163 subunit were performed, yielding a local map of 3.3 Å (Map H) and a local map of 3.4 Å (Map I), respectively. Particles corresponding to the best class covering the third CD163 subunit from the focused 3D classification were also re-imported to cryoSPARC and subjected to homogeneous, non-uniform refinement and local refinement with particle subtraction, resulting in a local map of 3.5 Å (Map J). Maps H to J were combined into a composite map (Map K) for model building.

For the CD163/HpSPHb dataset, 490,343 particles selected from the 2D classification were used to generate four ab initio models followed by heterogeneous refinement. A subset of 222,638 particles, displaying HpSPHb in complex

with three CD163 subunits, yielded a 3D reconstruction of 4.3 Å resolution after homogenous refinement. Similarly, the particles of this subset were exported to RELION and subjected to focused 3D classification without alignment with a mask around the third CD163 subunit or the rest of the complex. Particles corresponding to the best class covering the rest of the complex were re-imported to cryoSPARC and refined with homogeneous and non-uniform refinement, yielding a consensus map of the complex with a nominal resolution of 4.0 Å (Map L). Particles corresponding to the best class covering the third CD 163 subunit from the focused 3D classification were re-imported to cryoSPARC and subjected to homogeneous, non-uniform refinement and local refinement with particle subtraction, resulting in a local map of 4.1 Å (Map M). Maps L and M were combined into a composite map (Map N) for model building.

Simplified flowcharts for image processing of each dataset were summarized in S1, S4 and S5 Figs. It should be noted that the electron density maps from particle reconstructions corresponding to dimeric configurations of CD163 in both unliganded and Hp(1–1)Hb-bound structures showed weak densities for the third CD163 chain, observed only at a very low counter level, indicating its low occupancy. The two remaining CD163-I and CD163-II chains, with strong densities, displayed the same self- and ligand-binding modes as in the two trimeric structures.

### Model building, refinement, and validation

Model building was carried out by fitting the available crystal structures of human Hp(1–1)Hb (PDB: 4WJG) and HpSPHb (PDB: 4X0L), and the AlphaFold2 model of human CD163-ECD into the EM density maps using UCSF ChimeraX [38]. All the models were then manually adjusted in Coot [39]. The identification of $Ca^{+2}$ ions was guided by the following criteria: (1) distinct, well-defined density peaks in cryo-EM maps, surpassing a contour level of 8 sigma; (2) the local amino acid environment and coordination chemistry, primarily involving negatively charged Asp and Glu clusters, indicating their interaction with positively charged cations; and (3) the presence of $Ca^{+2}$ ions at 2–5 mM $CaCl_2$ in the protein buffer, consistent with the $Ca^{+2}$-binding properties of SRCR domains. The final model refinement was carried out using phenix.real_space_refine in PHENIX [40] with geometry restraints. The final atomic models were evaluated using MolProbity [41]. A summary of deposited maps, models and refinement statistics are provided in S1 and S2 Tables. Structural figures were prepared by UCSF ChimeraX [38].

### Supporting information

**S1 Fig. Cryo-EM analysis of the unliganded CD163. (A)** Flow chart of data processing. Details can be found in the Image processing section. **(B)** Representative cryo-EM micrograph. **(C)** Representative 2D class averages. **(D)** Local resolution estimation and orientation distribution plot for the consensus map A. **(E–H)** Local resolution estimation for the local refinement maps B–E. **(I)** Gold-standard FSC curves for the consensus maps A and local refinement maps B–E.
(TIFF)

**S1 Table. Cryo-EM data collection, refinement and validation statistics of CD163.**
(TIFF)

**S2 Fig. Structure of CD163 ECD and its calcium binding sites. (A)** Structure of CD163 ECD. The CD163 ECD structure was assembled by fitting the structure of domain D1, predicted by AlphaFold2, and the structure of domains D2–D9 from the unliganded CD163 model, to the map contoured at a low level. **(B)** Structures of SRCR domains of SCARA5 (PDB: 7C00) and CD163 (D2 to D9). Residues involved with cation-binding and $Ca^{+2}$ ions are shown as sticks and spheres, respectively. **(C)** $Ca^{+2}$ ions bound to the acidic clusters in CD163 D2, D3, D6, D7, and D9. Residues involving $Ca^{+2}$ binding and $Ca^{+2}$ ions bound to these acidic clusters are shown as sticks and spheres, respectively. The cryo-EM map in these regions is shown as gray surface. **(D)** Multiple sequence alignment of SRCR domains of SCARA5 and CD163 (D1 to D9). Conservation of the cation-binding residues are highlighted in red.
(TIFF)

**S3 Fig. Purification of CD163/Hp(1–1)Hb and CD163/HpSPHb complexes. (A)** Size-exclusion chromatogram and **(B)** corresponding SDS–PAGE analysis of the purified CD163/Hp(1–1)Hb complex. The fraction highlighted in the chromatogram was analyzed in the SDS–PAGE reducing gel showing each component of the complex. **(C)** Size-exclusion chromatogram and **(D)** corresponding SDS–PAGE analysis of the purified CD163/HpSPHb complex. The fraction highlighted in the chromatogram was analyzed in the SDS–PAGE reducing gel showing the components of the complex.
(TIFF)

**S4 Fig. Cryo-EM analysis of the CD163/Hp(1–1)Hb complex. (A)** Flow chart of data processing. Details can be found in the Image processing section. **(B)** Representative cryo-EM micrograph. **(C)** Representative 2D class averages. **(D)** Local resolution estimation and orientation distribution plot for the consensus map G. **(E–F)** Local resolution estimation for the local refinement maps H–I. **(G)** Local resolution estimation and orientation distribution plot for the local refinement map J. **(H)** Gold-standard FSC curves for the consensus maps G and local refinement maps H–J.
(TIFF)

**S2 Table. Cryo-EM data collection, refinement, and validation statistics of CD163/Hp(1–1)Hb and CD163/HpSPHb.**
(TIFF)

**S5 Fig. Cryo-EM analysis of the CD163/HpSPHb complex. (A)** Flow chart of data processing. Details can be found in the Image processing section. **(B)** Representative cryo-EM micrograph. **(C)** Representative 2D class averages. **(D)** Local resolution estimation and orientation distribution plot for the consensus map L. **(E)** Local resolution estimation and orientation distribution plot for the local refinement map M. **(F)** Gold-standard FSC curve for the consensus maps L and local refinement maps M.
(TIFF)

**S6 Fig. $Ca^{+2}$-mediated electrostatic interactions between CD163 subunits in the CD163/Hp(1–1)Hb complex. (A)** $Ca^{+2}$-dependent reciprocal electrostatic pairings between D7 and D9 of CD163 subunits. Key residues involved in the interactions and $Ca^{+2}$ ions bound to the acidic clusters in these interfaces are shown as sticks and spheres, respectively. Magenta dashed lines indicate salt bridges between Lys/Arg and Asp/Glu residues. Yellow dashed lines indicate hydrogen bonds. The cryo-EM map in these regions is shown as gray surface. **(B)** Schematic illustration of the SPR experiments examining binding of immobilized Hp(1–1)Hb to CD163(D1–D9) or CD163(D1–D5). **(C–D)** SPR sensorgrams of Hp(1–1)Hb binding to CD163(D1–D9) **(C)** and to CD163(D1–D5) **(D)**. Each data point is shown (25−2,000 nM, $n = 3$). The affinity between Hp(1–1)Hb and CD163(D1–D9) was estimated ($K_D = 72$ nM) using kinetic analysis mode. The affinity between Hp(1–1)Hb and CD163(D1–D5) was estimated ($K_D = 684$ nM) using equilibrium analysis mode because the binding was weak and could not be estimated using kinetic analysis mode. Source data for **(C–D)** can be found in S1 Data.
(TIFF)

**S1 Movie. Conformational changes of CD163 trimer upon Hp(1–1)Hb binding.** This movie was generated by morphing the structure of apo CD163 trimer and the structure of CD163/Hp(1–1)Hb complex, which were aligned using D6 to D8 of CD163-I. The structures are shown as cartoons and colored as in Figs 1 and 2.
(MP4)

**S7 Fig. Structural models of dimeric Hp(1–1)Hb and trimeric Hp(2–1)/Hb molecules bound by CD163 trimers on the cell surface. (A)** Structural model of a CD163 trimer on the cell-surface bound to Hp(1–1)/Hb. **(B)** Structural model of two CD163 trimers on the cell surface crosslinked by Hp(2–1)/Hb multimer. All model parts for which the structures are not available – the Hp(2–1)2 complex, the TM helices and the linkers connecting each C-terminus of D9 to the TM domain – were built using the multimer settings in AlphaFold protein structure prediction platform [25].
(TIFF)

**S1 Data.  Source data for graphs in this paper.**
(XLSX)

**S1 Raw Images.  Uncropped SDS–PAGE gel images in this paper.**
(PDF)

## Acknowledgments

Cryo-electron microscopy was performed at the Astbury Biostructure Laboratory, University of Leeds. Mass photometry was carried out at the Biomolecular Interactions Facility, Faculty of Biological Sciences, University of Leeds. The authors thank Mohammed Hussain for training on the Refeyn instrument and Iain Manfield for his valuable advice throughout the measurements.

## Author contributions

**Conceptualization:** Ching-Shin Huang, Hui Wang, Javier Chaparro-Riggers, Lidia Mosyak.

**Data curation:** Ching-Shin Huang, Joshua B. R. White, Oksana Degtjarik, Cindy Huynh, Kristoffer Brannstrom, Mark T. Horn.

**Formal analysis:** Ching-Shin Huang, Joshua B. R. White, Oksana Degtjarik, Cindy Huynh, Kristoffer Brannstrom, Mark T. Horn.

**Funding acquisition:** William S. Somers, Javier Chaparro-Riggers, Laura Lin, Lidia Mosyak.

**Investigation:** Ching-Shin Huang, Hui Wang, Joshua B. R. White, Oksana Degtjarik, Cindy Huynh, Kristoffer Brannstrom, Mark T. Horn, Lidia Mosyak.

**Methodology:** Ching-Shin Huang, Joshua B. R. White, Oksana Degtjarik, Cindy Huynh, Kristoffer Brannstrom, Mark T. Horn, Stephen P. Muench.

**Project administration:** Ching-Shin Huang, Hui Wang, Lidia Mosyak.

**Resources:** Ching-Shin Huang, Hui Wang, Stephen P. Muench, William S. Somers, Laura Lin, Lidia Mosyak.

**Software:** Ching-Shin Huang, Joshua B. R. White, Oksana Degtjarik.

**Supervision:** Ching-Shin Huang, Hui Wang, Stephen P. Muench, William S. Somers, Javier Chaparro-Riggers, Laura Lin, Lidia Mosyak.

**Validation:** Ching-Shin Huang.

**Visualization:** Ching-Shin Huang, Joshua B. R. White, Oksana Degtjarik.

**Writing – original draft:** Ching-Shin Huang, Lidia Mosyak.

**Writing – review & editing:** Ching-Shin Huang, Hui Wang, Joshua B. R. White, Oksana Degtjarik, Cindy Huynh, Kristoffer Brannstrom, Mark T. Horn, Stephen P. Muench, William S. Somers, Javier Chaparro-Riggers, Laura Lin, Lidia Mosyak.

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
