## [Editor Report · Decision Letter 0]

Dear Dr Huang, 

Thank you for submitting your manuscript entitled "Structural elucidation of the haptoglobin–hemoglobin clearance mechanism by macrophage scavenger receptor CD163" for consideration as a Research Article by PLOS Biology.

Your manuscript has now been evaluated by the PLOS Biology editorial staff, as well as by an academic editor with relevant expertise, and I am writing to let you know that we would like to send your submission out for external peer review. Given the previous publication, we will let the reviewers know that your study in under the scooping protection policy, so they do not consider that study to measure novelty. 

Once your full submission is complete, your paper will undergo a series of checks in preparation for peer review. After your manuscript has passed the checks it will be sent out for review. To provide the metadata for your submission, please Login to Editorial Manager (https://www.editorialmanager.com/pbiology) within two working days, i.e. by Feb 22 2025 11:59PM.

Kind regards,

Melissa 

Melissa Vázquez Hernández, PhD

Associate Editor, PLOS Biology

on behalf of 

Richard

Richard Hodge, PhD

Senior Editor

PLOS Biology

rhodge@plos.org

---

## [Decision Letter · Decision Letter 1]

Dear Ching-Shin,

Thank you for your continued patience while your manuscript "Structural elucidation of the haptoglobin–hemoglobin clearance mechanism by macrophage scavenger receptor CD163" was peer-reviewed at PLOS Biology. Please accept my sincere apologies for the delays that you have experienced during the peer review process. Your manuscript has now been evaluated by the PLOS Biology editors, an Academic Editor with relevant expertise, and by two independent reviewers. 

In light of the reviews, which you will find at the end of this email, we would like to invite you to revise the work to thoroughly address the reviewers' reports.

As you can see, both reviewers are very positive about the study and note it is well-structured and well done. They both raise several concerns that should be addressed in a revision, including discrepancies between the solved structure and the corresponding gels, clarifying the mechanistic model proposed and ruling out indirect EDTA effects using functional-separation mutants.

Given the extent of revision needed, we cannot make a decision about publication until we have seen the revised manuscript and your response to the reviewers' comments. Your revised manuscript is likely to be sent for further evaluation by all or a subset of the reviewers.

**IMPORTANT - SUBMITTING YOUR REVISION**

*Re-submission Checklist*

*Published Peer Review*

*PLOS Data Policy*

*Blot and Gel Data Policy*

Sincerely,

Richard

Richard Hodge, PhD

rhodge@plos.org

REVIEWS:

Reviewer #1: The scavenger receptor CD163 plays a critical role in cellular defense by mediating macrophage internalization of free hemoglobin (Hb) during intravascular hemolysis, thereby mitigating vascular and renal tissue damage. In this study, the authors provide critical structural and mechanistic insights into CD163's cargo recognition process by resolving its ligand-free (3.0 Å) and ligand-bound (3.3 Å) conformations. Key findings include the identification of an asymmetric, autoinhibitory trimeric state of unliganded CD163, which sterically blocks the binding site for the haptoglobin-hemoglobin (HpHb) heterodimer. Upon HpHb binding, the receptor transitions to a near-symmetrical 3:1 CD163/HpHb complex. The authors further demonstrate that calcium ions, acting at subunit interfaces, are indispensable for stabilizing CD163 self-assembly, as EDTA-mediated Ca²⁺ chelation disrupts trimerization and reverts the receptor to a monomeric state. This calcium-dependent network is proposed to regulate conformational switching between ligand-free and ligand-bound states, offering a compelling mechanistic framework for the spatiotemporal control of HpHb clearance.

The manuscript is well-structured, methodologically rigorous, and addresses a biologically important question. The conclusions are supported by the data, and the comparative analysis with previous structural work strengthens the novelty of the findings. This study is suitable for publication in PLOS Biology following minor revisions to address the following points: 

- The role of Ca2+ in the CD163/HpHb complex assembly is demonstrated through the bulk removal of Ca2+ by EDTA. However, the authors already showed that EDTA treatment could dissemble the unliganded CD163 trimer into monomers. Structure-guided functional-separation mutants would strengthen the mechanistic claims and rule out indirect EDTA effects.

- Panels b-d in Fig.4 are visually overcrowded. The authors may simplify the presentation by highlighting a single subunit with sharper color contrast. 

- "Hp(1-1)Hp" should be "Hp(1-1)Hb" in Fig.4. legend. 

Reviewer #2: The study by Huang et al. reports the structure and mechanism of CD163's binding and import of the haptoglobin-hemoglobin complex. CD163 is a type 1 membrane protein responsible for binding and clearing the circulating HpHb complex. This work resolves the CryoEM structure of CD163 alone and bound to the HpHb complex. These results reveal a significant structural change between the bound and apo states, where the binding site is sequestered in the absence of substrate. These structural changes appear highly dependent upon structural calcium ions at the CD163:CD163 and CD163:HpHb interfaces. 

The study is carefully designed, thorough, and well carried out. Generally, the results are properly analyzed and the scientific conclusions are clearly presented and well justified. There is no question the paper should be published. However, I have a few modest points regarding interpretation of their data and the mechanistic model.

Major concerns:

- The authors have clearly resolved the CD163/HpSPHb complex from their CryoEM dataset. However, the corresponding gel for this sample (Supp. Fig 3d) appears to show only sub-stoichometric amounts HpSP and Hb in the peak fraction, relative to CD163. How do the authors explain this discrepancy? 

- Similarly in Supp Fig 3b, there appears to be two bands in the Hp1-1 samples, and both are carried over into the complex peak. Is this additional component identified in the sample and/or apparent in the CryoEM dataset?

- The clash socre in the CD163/HpSPHb structure appears excessively high, particularly in light of the modest resolution. This should be re-refined with optimized restraints.

- How the Ca+2 ions were identified is not explained. A justification for modeling these unknown densities as calcium, relative to other possible ions, should be included. 

- The mechanism for HpHb binding by CD163 is not entirely clear from the author's model and results. If several calcium ions stabilize both the auto-inhibited and bound state, how does this transition occur? Do they need to be released and then re-bind in the conformational change, and does this have consequences for the calcium-dependence of this structural transition. 

- Figure 4 panels B, C, and D, the colors are far too pale and therefore not visible on a screen or printed. These should be revised

Minor concerns:

- The color for salt bridges in Figure 3 is too close to the colors for protein and density and therefore difficult to see. We suggest changing this.

Trivial concerns:

- The authors occasionally mix single-letter and three-letter amino acid codes, even within the same sentence (such as lines 158-160). This is particularly confusing as the CD163 domains are also described using a single-letter and number code. Therefore, a three-letter code should be used throughout for clarity.

---

## [Editor Report · Decision Letter 2]

Dear Dr Huang,

Thank you for your patience while we considered your revised manuscript "Structural elucidation of the haptoglobin–hemoglobin clearance mechanism by macrophage scavenger receptor CD163" for publication as a Research Article at PLOS Biology. This revised version of your manuscript has been evaluated by the PLOS Biology editors and the Academic Editor.

Based on our Academic Editor's assessment of your revision, I am pleased to say that we are likely to accept this manuscript for publication, provided you satisfactorily address the following data and other policy-related requests that I have provided below (A-F):

(A) You may be aware of the PLOS Data Policy, which requires that all data be made available without restriction: http://journals.plos.org/plosbiology/s/data-availability. For more information, please also see this editorial: http://dx.doi.org/10.1371/journal.pbio.1001797

-Supplementary files (e.g., excel). Please ensure that all data files are uploaded as 'Supporting Information' and are invariably referred to (in the manuscript, figure legends, and the Description field when uploading your files) using the following format verbatim: S1 Data, S2 Data, etc. Multiple panels of a single or even several figures can be included as multiple sheets in one excel file that is saved using exactly the following convention: S1_Data.xlsx (using an underscore).

-Deposition in a publicly available repository. Please also provide the accession code or a reviewer link so that we may view your data before publication. 

Figure 1G-H, 3G-H, S8C-D

(B) Thank you for providing the structural data in the PDB and EMDB databases. However, we note that the data is currently on hold for release. We ask that you please make the structures publicly available at this stage before publication.

(C) Please also ensure that each of the relevant figure legends in your manuscript include information on *WHERE THE UNDERLYING DATA CAN BE FOUND*, and ensure your supplemental data file/s has a legend.

(D) We require the original, uncropped and minimally adjusted images supporting all blot and gel results reported in the following Figures:

Figure S4B, S4D

We will require these files before a manuscript can be accepted so please prepare and upload them now. Please carefully read our guidelines for how to prepare and upload this data: https://journals.plos.org/plosbiology/s/figures#loc-blot-and-gel-reporting-requirements

(E) Per journal policy, if you have generated any custom code during the course of this investigation, please make it available without restrictions. Please ensure that the code is sufficiently well documented and reusable, and that your Data Statement in the Editorial Manager submission system accurately describes where your code can be found. 

(F) Please note that per journal policy, the model system/species studied should be clearly stated in the abstract of your manuscript. 

We expect to receive your revised manuscript within two weeks. 

*Published Peer Review History*

*Press*

Best regards,

Richard

Richard Hodge, PhD

rhodge@plos.org

PLOS

---

## [Editor Report · Decision Letter 3]

Dear Ching-Shin,

On behalf of my colleagues and the Academic Editor, Yan Zhang, I am pleased to say that we can accept your manuscript for publication, provided you address any remaining formatting and reporting issues. These will be detailed in an email you should receive within 2-3 business days from our colleagues in the journal operations team; no action is required from you until then. Please note that we will not be able to formally accept your manuscript and schedule it for publication until you have completed any requested changes.

PRESS

Best wishes, 

Richard

Richard Hodge, PhD

rhodge@plos.org

PLOS
